# Monothiooxalamide–Benzothiazole Hybrids: Predictive Docking on HDAC6, Synthesis, Molecular Structure, and Antiproliferative Activity on Breast Cancer Cells

**DOI:** 10.3390/ijms26178684

**Published:** 2025-09-05

**Authors:** Carlos Eduardo Macías-Hernández, Irving Balbuena-Rebolledo, Efrén V. García-Báez, Laura C. Cabrera-Pérez, Marycarmen Godínez-Victoria, Martha C. Rosales-Hernández, Itzia I. Padilla-Martínez

**Affiliations:** 1Laboratorio de Química Supramolecular y Nanociencias, Unidad Profesional Interdisciplinaria de Biotecnología, Instituto Politécnico Nacional, Avenida Acueducto s/n, Barrio la Laguna Ticomán, Mexico City 07340, Mexico; macias.hernandez.94@gmail.com (C.E.M.-H.);; 2Laboratorio de Citometría de Flujo e Inmunología Clínica, Sección de Estudios de Posgrado e Investigación, Escuela Superior de Medicina, Instituto Politécnico Nacional, Plan de San Luis y Salvador Díaz Mirón s/n Casco de Santo Tomas, Mexico City 11340, Mexico; maric_27@yahoo.com; 3Laboratorio de Biofísica y Biocatálisis, Sección de Estudios de Posgrado e Investigación, Escuela Superior de Medicina, Instituto Politécnico Nacional, Plan de San Luis y Salvador Díaz Mirón s/n, Casco de Santo Tomas, Mexico City 11340, Mexico

**Keywords:** benzothiazole, monothiooxalamide, antiproliferative, HDAC6

## Abstract

A new family of monothiooxalamide derived from 2-aminobenzothiazole was synthesized with the purpose of investigating its anticancer activity. The design of the compounds was focused on targeting the HDAC6 enzyme, a target for antineoplastic drugs. The in silico affinity of compounds to HDAC6 was performed and confirmed by docking simulation. The structures of monothiooxalamide–benzothiazole hybrids were characterized by 1D and 2D NMR experiments, as well as through mass spectrometry and IR spectroscopy. In addition, the antiproliferative activity of compounds was assessed in human breast cancer cell lines (MCF-7 and MDA-MB231) and non-malignant cells (MCF-10A and NIH/3T3). The most active compound was *N*-(benzo[d]thiazol-2-yl)-2-((4-methoxybenzyl)amino)-2-thioxoacetamide (**1c**), which inhibited breast cancer cell growth and invasiveness in vitro and induced late apoptosis in the MCF-7 cell line. The molecular structure of **1c** was solved by single-crystal X-ray diffraction. The supramolecular arrangement of benzothiazole and 4-methoxy-benzylamine moieties, present in the crystal structure of **1c**, was consistent with the interactions on the docked DD2-HDAC6 catalytic site.

## 1. Introduction

Monothiooxalamides (MTAs) are a group of oxalic acid derivatives, analogous to compounds named as oxalamides and thiooxalamides, Figure 1. The ability of these compounds to form intramolecular hydrogen bonds has been extensively studied [1,2]. It is well known that intramolecular and intermolecular hydrogen bonding exerts a strong influence on the physicochemical properties of compounds, such as their crystalline structure and solubility. Moreover, it can also be critical for the interaction between small molecules and many biological targets [2,3]. The stability provided by intramolecular hydrogen bonds in MTA and oxalamides has been previously reported, allowing these compounds to be stored for extended periods, a desirable property in the pharmaceutical industry [2].

Nitrogen-sulfur-containing heterocyclic compounds such as thiazole and benzothiazole (BT) are biologically important building blocks in drug discovery and development [4,5,6,7]. They are widely distributed in several commercially available drugs and natural products [8,9]. The BT fragment is present in compounds that exhibit a wide range of biological activities, including anticancer [10,11,12,13,14,15], antimicrobial [13,16], and antifungal [13] effects. Particularly, 2-aminobezothiazole (2ABT) derivatives have deserved great interest due to its antitumor activity [17]. Many biological targets have been identified for this heterocycle, such as tyrosine-kinase receptors (RTKs), which are involved in crucial processes of cell proliferation, cell cycle, metabolism, and cell migration [8,18]. Other targets of interest in cancer therapy are Histone Deacetylase enzymes (HDACs), a group of cancer-related proteins involved in tumorigenesis and cancer progression, particularly the isoforms 4, 6, and 8. These HDACs are highly expressed in MCF-7 and MDA-MB-231 cells and play a critical role during cancer cell proliferation [19,20,21,22,23]. Numerous 2ABT derivatives exhibit HDAC inhibitory activity, demonstrating that this heterocycle could be a suitable alternative for the development of anticancer drugs [8]. On the other hand, MTAs have a planar conformation due to their inherent intramolecular hydrogen bonding capabilities [2]. It is known that planar structures are effective DNA intercalating agents, as they can induce changes in DNA conformation, ultimately leading to the death of cancer cells. In continuation with our studies on the antiproliferative activity of MTAs [24], this work reports the synthesis of novel MTAs with a 2ABT moiety, named MTA-BTs, whose antiproliferative capacity was assessed, targeting the HDAC6 enzyme.

## 2. Results and Discussion

### 2.1. Rational Design of Compounds

The classical HDAC6 inhibitors consist of a tripartite structure. It contains a capping group that usually occupies the rim of the pocket (L1 or L2 domains), followed by a linker that interacts with the hydrophobic tunnel of the catalytic domain, and a zinc-binding group that interacts with the metal ion [25,26,27,28,29,30,31]. Typically, the zinc-binding group is a hydroxamate that efficiently interacts with zinc in the catalytic pocket. However, it has been demonstrated that hydroxamate is susceptible to glucuronidation reactions by metabolism, considerably reducing the bioavailability of the drug [32,33,34]. Furthermore, the hydroxamate group has also been associated with mutagenicity in belinostat, vorinostat, and panobinostat [35,36,37]. The importance of improving these characteristics of classical HDAC6 inhibitors could be crucial to achieving better results in antineoplastic pharmacological strategies and reducing their side effects. In addition, it is well known that the hydrophobic tunnel is wider in HDAC6 than in the other isoforms. This leads to the hypothesis that the addition of bulky aromatic groups to the structure could enhance the interaction with the hydrophobic channel, thereby improving the affinity with the enzyme. In this context, we proposed the design of an MTA family of compounds containing a BT group, which has also been associated with exhibiting a wide range of biological activities, such as anticancer and antimicrobial [8,38,39,40]. The MTA group was proposed as a linker to improve the interaction of compounds with the hydrophobic tunnel. Additionally, an aromatic ring was incorporated into the design to enhance interaction with the wide hydrophobic channel of HDAC6, Figure 2.

### 2.2. Synthesis and NMR Assignments

MTA-BTs **1a**–**d** and **2a**–**d** were obtained by the inexpensive method previously established by our research group [2,24,41] (Figure 1), using a chloroacetamide as an intermediate (**1** and **2**), the appropriate amine (**a**–**d**), and S_8_ as a sulfur source. Compounds **1a**,**b**, and **2a** were purified by recrystallization from ethanol, in medium to good yields (63–76%). Whereas **1c**,**d**, and **2b**–**d** were purified by column chromatography in medium yields (38–51%).

Both families of MTA-BTs were characterized by ^1^H, ^13^C, and 2D NMR experiments. The results confirmed the structures of the compounds. The characteristic high-frequency chemical shifts of the amide (δN_O_H) and thioamide (δN_S_H) protons were observed in the ^1^H NMR spectra of compounds **1a**–**d** and **2a**–**d**. δN_O_H is in the 11.22–11.31 ppm range, whereas the δN_S_H is in the 9.38–9.77 ppm range. The COSY experiment confirmed the correct assignments of both signals, Figure 3. The N_O_H appears as the most deshielded signal, probably due to intramolecular hydrogen bonding engagement with the thiocarbonyl sulfur (CS), as previously reported by our research group [2,41]. Additionally, the spin-spin splitting between N_S_H (H13) and the neighboring H14 protons was observed in all compounds (^3^*J_13-14_ =* 5.6–6.4 Hz). This result suggests the stabilization of the N_S_H by intramolecular hydrogen bonding with the amide carbonyl (CO). It is worth highlighting that the thioamide proton is usually engaged in fast acidic hydrogen exchange [2,41].

In the ^13^C-spectra, the characteristic CS group appears in the 182.3–184.5 ppm range, and the CO signal appears in the 156.1–160.8 ppm range. The deshielding effect observed on the CS is stronger than on CO; this effect is explained because of the preferred delocalization of the lone pair of thioamide nitrogen (N_CS_) to the CS group, in contrast to the shared delocalization of the lone pair of amide nitrogen (N_CO_) to both the CO group and the BT ring [42].

### 2.3. Molecular Structure and Intramolecular Hydrogen Bonding of Compound ***1c***

NMR is a widely used technique to elucidate the structure of organic compounds. However, single-crystal X-ray diffraction is one of the most accurate methods for determining the molecular structure of compounds, as well as their supramolecular arrangement. A detailed description of the three-dimensional structure of compounds is essential in drug design because it leads to a better understanding of the kind of supramolecular interactions that compounds can establish in the crystal structure, which can be correlated with those interactions that could be established between compounds and biological targets [43,44,45,46]. Compound **1c** crystallized as a triclinic crystal system in the space group *P-1*, with two molecules in the unit cell. Details of the crystal data, data collection, and structure refinement are summarized in Appendix A. The molecular structure of **1c** is displayed in Figure 4. The experimental bond lengths values of **1c** are in agreement with aromatic C—X (X = C, N, S), C=O and C=S reported values [47], and therefore with two isolated CO and CS groups (C=O = 1.234(12) Å, C=S = 1.681(20) Å), as has been found in phenylene bis-monothiooxalamides [2]. The complete list of parameters inherent to the geometry of compound **1c** is given in the Appendix A.

N_O_H and N_S_H are anti-positioned between each other, with YCNH torsion angle values of 179(2)° (Y = O11) and 179(2)° (Y = S12), and O11C11C12S12 torsion angle of 178.7(2)°. The thiooxamide group is almost coplanar with the BT heterocycle, forming an angle of only 4(2)° between the two planes; however, it is tilted away from the (4-methoxy)benzylamine residue by 77(2)°. Nevertheless, the lone pair of the N_CS_ is more delocalized to the CS (N13—C12 = 1.316(4) Å) than the lone pair of the N_CO_ to the CO (N10—C11 = 1.351(4) Å).

Thiooxamide group is intramolecularly hydrogen bonded, displaying the typical pattern of adjacent S(5)S(5)S(5) set of rings formed by NH···Y (X = O11, S12) and CH···S12 interactions. The geometric parameters associated with intramolecular hydrogen bonding are listed in Table 1.

In addition to coplanarity between BT-heterocycle and thiooxamide group, the heterocyclic sulfur S1 atom and the carbonyl oxygen O11 are in *syn* conformation at a distance of 2.808(3) Å, below the sum of the van der Waals radii (r_vdW_ = 3.32 Å) [48], and a C8—S1⋯O11 angle of 161.6(1)°, in agreement with chalcogen bonding (n_O_ → σ*_C–S_) [49], forming an *S(5)* ring motif. The interaction distance O11⋯S1 and C8―S1⋯O11 angle values agree with reported values [50].

### 2.4. Supramolecular Structure of Compound ***1c***

In the Hirshfeld surface (HS) of compound **1c**, the strongest interaction (O⋯H/H⋯O) is given by N—H13⋯O11 Hydrogen Bond (HB), Figure 5a, which is shown as bright red spots. Among all the close interactions, H⋯H contacts represent the most extensive set of interactions contributing to the surface, accounting for 39.6%. In addition, the H⋯A/A⋯H (A = C, N, O, S) interactions, represent the 45.6% as a whole, whereas the remaining 14.8% is given by C⋯X/X⋯C and N⋯X/X⋯N (X = C, N, O, S) contacts, Figure 5b.

The geometric parameters associated with intermolecular interactions are listed in Table 1. Selected results of quantitative interaction energy analysis of **1c**, performed at the B3LYP/6-311G(d,p) level of theory, are tabulated in Table 2. Interaction energy (E_tot_) values smaller than −20.0 kJ mol^−1^ were neglected. The most energetic interaction, N—H13⋯O11 hydrogen bonding, assembles as *R^2^_2_(10)* intermolecular ring motif, whose geometric parameters agree with a strong interaction. In fact, this motif contributes with −63.3 kJ mol^−1^ to the energy in the crystal and shapes a centrosymmetric dimer, Figure 6a. The nature of the attractive energy involved in N—H13⋯O11 hydrogen bonding (HB) is primarily electrostatic, with a percent contribution to the stabilization energy (%E_ele_) of 60.4%.

The second largest energy contributor to the crystal energy, with −59.0 kJ mol^−1^, is the π-assembly between two thiazole rings, with distances of 3.6317(17) Å and 3.5301(12) Å between *Cg1*⋯*Cg1* centroids and *Cg1* perpendicular planes, respectively (*Cg1* is the centroid of the S1C2N3C9C8 ring; symmetry code = 1-x, 1-y, 1-z) (Figure 6b). This π-stacking interaction is mainly dispersive, with a %E_dis_ value of 80.0%.

At last, C12=S12⋯*Cg2* interaction, of n_S_ → π* type, contributes with −40.1 kJ mol^−1^, being mostly dispersive, with %E_dis_ value of 89.0% of its attractive energy, Table 2. Both *Cg1*⋯*Cg1* and S12⋯*Cg2* interactions between centrosymmetric π-stacked dimers develop the second dimension (2D) in the (0 4 7) crystal plane, Figure 6b. Thus, the contribution of soft π-stacking interactions (E_tot_ = −99.1 kJ mol^−1^) to the stability of the crystal network of **1c** is the largest and mostly dispersive in nature.

The visual representation of the energy-framework diagrams for E_ele_ (red), E_dis_ (green) and E_tot_ (blue) for a cluster of nearest-neighbor molecules is shown in Figure 7. The crystal network of **1c** is mainly dominated by the E_dis_ component, showing zig-zagging energy framework, Figure 7b, that overcome the E_elec_ component given by the most energetic N—H13⋯O11 interaction, Figure 7a, to provide the final shape to the E_tot_ energy framework, Figure 7c.

### 2.5. Molecular Docking to DD2-HDAC6 Catalytic Site

The affinity of MTA-BT derivatives **1a**–**d** and **2a**–**d** for the DD2-HDAC6 catalytic site was evaluated by performing a docking simulation. The results are expressed in terms of free energy of ligand-receptor binding (ΔG^o^_b_) and dissociative equilibrium constant (K_d_), Table 3. According to the docking calculations, all compounds interact with the DD2-HDAC6 catalytic site, showing ΔG^o^_b_ values ranging from −7.3 to −8.5 kcal/mol. The values were close to Nexturastat A (NA) (ΔG^o^_b_ = −7.8 kcal/mol), the reference drug and the crystallized ligand used for docking validation. The binding modes of MTA-BTs and NA revealed that the compounds reached the catalytic DD2 domain of HDAC6, as shown in Figure 8. The tested compounds interact with the surface binding domain of the catalytic site through the BT system and the thiooxalamide group, while the aromatic ring, from the amine residue, is introduced into the hydrophobic channel, efficiently blocking the catalytic site. The presence of the phenethyl group in the R^2^ position of compounds **1d** and **2d** improves the interaction with the hydrophobic channel by π-dispersive forces with Phe-544, Phe-604, and His-575. For compounds **1c** and **2c**, the *para*-methoxy group on the aromatic ring (R^2^) appears to reinforce the interaction with the hydrophobic channel by binding to His-535 and His-575 through π-alkyl interactions, thereby blocking access to the catalytic site of DD2-HDAC6. On the other hand, for compounds **1b** and **2b**, with an *ortho*-methoxy group, the effect of the substituent makes the interaction with the hydrophobic channel slightly less favorable. Regarding the reference drug, the binding mode of the NA with the DD2-HDAC6 catalytic site shows that the ligand reaches the catalytic site domain by blocking the surface (His-424 and Pro-425) through phenyl and butyl groups (Phe-604). A strong binding with the hydrophobic channel is also observed. To achieve a good blocking of the hydrophobic channel, it is crucial to bind the aromatic rings with Phe-604, Phe-544, and His-575 by π-interactions. In addition, the molecular structure of compound **1c**, which was determined by single-crystal X-ray diffraction, has an almost 80° angle in which the (4-methoxy)benzylamine residue is involved, as shown in Figure 4. This arrangement is induced by hydrogen bonds between H10, S12, and H14 atoms and π-interactions between the BT rings, Figure 6. Taking into account the tridimensional structure of the DD2-HDAC6 catalytic domain, in which the pocket rim and hydrophobic channel are involved on HDAC6 inhibition, the planar structure of BT moiety on the crystal structure seems to contribute with the affinity against the rim of the pocket by π-cation interactions with His-424 and Asn-418, similarly with occurs on crystal structure between BT rings. Additionally, the planar structure of the (4-methoxy)benzylamine moiety reinforces the interaction with the hydrophobic tunnel through π-π interactions with Phe-544 and Phe-604, resulting in a good cytotoxic effect exerted by **1c** on human cancer cell lines [26,51,52]. NA can also interact with the zinc-binding domain by the hydroxamate group. This interaction involves hydrogen bonding with His-534 and His-535. It is worth noting that this kind of interaction, which involves the surface, the hydrophobic channel, and the zinc-binding domains, was also previously described as crucial to efficiently inhibit the catalytic activity of HDAC6 [23,53,54,55,56]. The complete list of interactions of NA, **1a**–**d**, and **2a**–**d** with DD2-HDAC6 is in the Appendix A.

### 2.6. Antiproliferative Activity

The antiproliferative effect of monothiooxalamides **1a**–**d** and **2a**–**d** was assessed in MCF-7 (human breast cancer cell line with estrogen, progesterone, and glucocorticoid receptors), MDA-MB-231 (triple-negative human breast cancer), MCF-10A (non-tumorigenic human mammary epithelial cell line), and NIH/3T3 (mouse fibroblast). The cytotoxicity is expressed as IC_50_ values in micromolar concentrations, Table 4. The most active compound was **1c** for both carcinogenic cell lines, with higher activity observed in MCF-7 cells. On the other hand, MDA-MB-231 cells were less sensitive to **1c** (1.6-fold more active on MCF-7 cells). Regarding non-tumorigenic MCF-10A cells, compound **1c** showed moderate selectivity toward malignant cells (1.7-fold more active in MCF-7 cells). In contrast, the cytotoxic effect on triple-negative cells MDA-MB-231 was similar to that observed in MCF-10A. In the other non-tumorigenic cell line, NIH/3T3, all compounds and MTX showed considerably more toxicity than both tumorigenic cell lines (MCF-7 and MDA-MB-231) and non-tumorigenic MCF-10A cells. Nevertheless, it is worth considering that NIH/3T3 are mouse fibroblasts, while the other three cell lines are human cells; therefore, the physiological differences between both species could play an essential role in the exerted effect. Several studies have evaluated 2-aminobenzothiazole derivatives against various cell lines. The targets of this kind of compounds are wide, including HDAC6, tyrosine kinase receptors family (CSF1R kinase, EGFR kinase, VEGFR-2 kinase, FAK kinase), serine/threonine kinases, lysine-specific demethylase (LSD1), phosphoinositide 3-kinase (PI3K), topoisomerases (I and II), Heat shock protein (HSP90), p53, nuclear receptor binding set (NSD1), microsomal prostaglandin E synthase-1 (mPGES-1), and also as chelators of metal ions, which are necessaries for critical biological processes such as gene expression, oxygen transport, and cell respiration [8]. In all cases, the IC_50_ values were reported in a wide range, from 1 nM to 50 µM. On the other hand, our research group reported a family of MTAs containing 2-aminopyridine, with an activity range of 0.19–18 µM against several human cancer cell lines (A2780-wt, A2780-cis-MSTO-211H, and HT-29) [24]. The tested compounds **1a**–**d** and **2a**–**d** fall within a medium range, between those reported previously for this kind of compounds (32–95 µM), with **1c** being the most active against MCF-7 and MDA-MB-231cancer cell lines. Although the tested compounds did not show high cytotoxicity against human cancer cell lines, the values are in the reported average for other BT compounds [17,57], and reported Re^I^-MTA complexes [58]. This study led to the identification of the most promising compound, **1c**, in which the presence of the 2-methoxyparaminoethyl structural feature appears to enhance antineoplastic activity significantly. Whereas **1c** could be a feasible candidate to obtain better results compared with **1a**, **2a**–**c**, which were highly cytotoxic to non-tumorigenic cells. As a perspective, the synergic activity of **1c**, in combination with antioxidants like curcumin, resveratrol, and quercetin, could also be explored [59,60,61], with the aim to sensitize cancer cells against cytotoxic agents and diminish their toxicity in non-tumorigenic cells. In contrast, MTX, which is an antineoplastic drug, is widely used for many kinds of cancer treatments, including breast cancer. Their mechanism of action is related to the inhibition of dihydrofolate reductase (DHFR). However, MTX shares some structural similarities with HDAC inhibitors, and some studies have demonstrated the ability of this drug to inhibit HDAC6 enzyme [62,63]. This drug was used as a reference and proved to be the most active against both tumorigenic cell lines (MCF-7 and MDA-MB-231). Nevertheless, MTX exerted an undesirable effect on NIH/3T3 cells. Although the compounds were not better than MTX, their antiproliferative effect was similar to 2ABT, thioureas, 3-carboxy-coumarin sulfonamides, pyrimethamine conjugates, and N-hydroxycinnamide analogs [38,64,65,66]. Highlighting its low toxicity to non-tumor cells, **1c** might be a better option than methotrexate to reduce adverse side effects associated with chemotherapy.

### 2.7. Cell Migration Assay

The triple-negative cancer, produced by MDA-MB-231 cells, is highly invasive and metastatic, capable of migrating to other tissues. To investigate the effect of MTA-BTs on inhibiting cell migration, a wound-closure assay was performed using MDA-MB-231 cells with the most active compound, **1c**. Cells were treated at the IC_50_ concentration (52 μM), and the wound closure was monitored at 0 h, 12 h, 24 h, and 48 h. The results are expressed as a percentage of the remaining wound surface after the wound, as shown in Figure 9. The wound closure begins at 12 h, at this time, there is a significant difference between cells treated with compound **1c** and MTX compared with non-treated cells. Nevertheless, the maximum effect is observed at 48 h, when wound closure is complete for non-treated cells. For cells treated with compound **1c**, the maximum wound closure was 51%, while for the reference drug, MTX, it was 38% at 48 h. In both cases, there is a significant difference from the control, with wound closure rates of −2.0 ± 0.1 (R = 0.98), −1.0 ± 0.1 (R = 0.94), and −0.7 ± 0.1 (0.88) in %area/h for control, **1c**, and MTX, respectively. The results confirmed the capability of **1c** to prevent cell migration in MDA-MB-231 cells, similar to the reference drug MTX, and other reported BTs [67,68].

### 2.8. Apoptosis Induction

Apoptosis induction is one of the most common mechanisms of cell death exerted by antineoplastic drugs. This is because cancer cells can evade this cell death mechanism, allowing malignant cells to survive and leading to tumor progression [69]. Apoptosis involves several physiological cell modifications, including the destabilization of the phospholipid bilayer and the subsequent exposure of phosphatidylserine. This phospholipid can be easily recognized by annexin V, a protein that is useful for identifying apoptotic cells [70]. Therefore, apoptosis was determined for both breast cancer human cell lines, MCF-7 and MDA-MB-231, and the NIH/3T3 non-tumoral cell line. The cells were treated with the most active compound **1c** and MTX, this last as a death-positive control, for 48 h of exposure. The viability of MCF-7 cells significantly decreased when treated with compound **1c** (55%) and MTX (39%); consequently, a significant increase in the late apoptosis cell population was observed for both treatments, as shown in Figure 10. This result was expected because MCF-7 cells were the most sensitive to **1c**. On triple-negative breast cancer cells (MDA-MB-231), methotrexate and **1c** did not show a significant effect on cell viability; nevertheless, MTX, the reference drug, appears to decrease the percentage of late apoptotic cells (9%) compared with non-treated cells (14%), as shown in Figure 11. Regarding non-malignant fibroblast cells (NIH/3T3), a slight decrease in live cells was observed only for compound **1c** (60%) compared to the control (81%) and MTX (72%), as shown in Figure 12.

### 2.9. Physicochemical and Toxicological Properties Prediction

The physicochemical and toxicological properties of MTA-BTs were predicted with OSIRIS Property Explorer; the results are listed in Appendix A. The n-octanol/water partition coefficient (cLog*p*) describes the hydrophobicity of compounds, which is crucial to cross biological membranes. cLog*p* values must be less than 5 to achieve a good absorption process in living organisms [41,71]. All compounds have cLog*p* values between 2.66 and 3.51. Log*S* is a parameter that expresses the solubility of the drugs. This physicochemical property helps predict drug absorption [41,72]. The Log*S* values must be greater than −4 to achieve a good absorption, distribution, and elimination process. In contrast, all compounds have values below −4, suggesting that aqueous solubility could be restricted. Nevertheless, it is worth noting that numerous drugs on the market have poor aqueous solubility. Regarding Log*S*, the best compounds are **1a, 1b**, and **1c.** Moreover, the toxicity risks of MTA-BTs **1a**–**d** and **2a**–**d** were also estimated, showing a low risk of toxicological effects; thus, both families of compounds are suitable in terms of toxicity.

The drug score expresses the potential of compounds to be feasible as a drug. The value combines physicochemical parameters and toxicity risk to obtain a global score (between 0 and 1), in which 1 is the best score and, 0 is the worst. Osiris has been widely used to select suitable compounds [73,74,75,76]. In a previous study, the toxicity of a series of benzimidazole derivatives on red blood cells was reported, and the prediction by OSIRIS was consistent with the in vitro evaluation [41]. In this study, compounds **1a**, **1b**, **1c**, and **2d** have the best drug-score values. Considering that compound **1c** is the most active compound against human cancer cell lines, particularly MCF-7 cells, it suggests that **1c** could be a promising drug candidate.

## 3. Materials and Methods

### 3.1. Materials and Equipment

All reagents were purchased from commercial suppliers and used without further purification. Chloroacetyl chloride, 2-aminobenzothiazole, phenethylamine, benzylamine, 2-methoxybenzylamine, 4-methoxybenzylamine, deuterated chloroform (CDCl_3_), deuterated DMSO (DMSO-d6), hexane, ethyl acetate, dichloromethane, sodium carbonate, potassium carbonate, and triethylamine. The solvents for column chromatography were distilled before use (hexane and ethyl acetate). Chloroacetamides **1** and **2** were obtained by the previously described method by our research group [2,24].

Melting points were measured using an Electrothermal Mel-Temp melting point apparatus (Cole-Parmer) and are uncorrected. ^1^H and ^13^C NMR spectra, were acquired on Varian NMR spectrometer Mercury-300 using CDCl_3_ or DMSO-d6 as solvent. All chemical shift values (δ) are reported in parts per million (ppm), using TMS as internal reference, and coupling constants (^n^*J*) are in Hz. The following abbreviations are used to indicate the multiplicity of the signals: s for a singlet, d for a doublet, t for a triplet, and q for a quadruplet. The numbering of compounds is shown in Appendix A and the NMR spectra are in Appendix A. Mass spectra were acquired using a micrOTOF-Q Bruker Daltonics instrument from acetonitrile solutions, Appendix A. FT-IR spectra were recorded neat at 25 °C using a PerkinElmer FT-IR Spectrum Two, equipped with a Miracle ATR Single Reflection Diamond (PIKE Technologies) device, and are reported in terms of wavenumber of absorption (cm^−1^), Appendix A. To indicate the intensity of the signals, the following abbreviations are used: w for weak, m for medium, and s for strong. The purity of the compounds was determined using an Agilent Technologies 1260 Infinity HPLC instrument (Santa Clara, CA, USA), using a ZORBAX XDC-C18 column, at 25 °C; HPLC chromatograms are shown in Appendix A.

### 3.2. General Procedure for the Synthesis of MTA-BTs ***1a–d*** and ***2a–d***

In a 100 mL ball flask, 2.50 mmol of the appropriate amine **a**–**d**, 10 mL of THF, 1 mL of triethylamine, and 1.20 mmol of elemental sulfur (0.311 g) were placed. The mixture was stirred at room temperature for 30 min. Subsequently, 2.30 mmol of chloroacetamide **1** (0.515 g) or **2** (0.547 g) was slowly added, and the reaction solution was stirred for 72 h at RT. Compounds **1a** and **1b** were purified by recrystallization from ethanol, and MTA-BTs by column chromatography using a hexane/ethyl acetate mixture of suitable polarity as eluent. All compounds were obtained as yellow solids and were spectroscopically characterized by 1D and 2D NMR (^1^H and ^13^C NMR), and IR.

*N-(benzo[d]thiazol-2-yl)-2-(benzylamino)-2-thioxoacetamide (***1a***)*. Obtained from 0.268 g of amine **a** and chloroacetamide **1** in 76% yield (0.562 g), m.p. 163–164 °C after recrystallization from ethanol. IR (cm^−1^) ν: 3267 (m, N-H amide I), N-H 3250 (br, thioamide I), 1683 (s, C=O), 1530 (m, N-H + C-N amide II), 1269 (s, C=S). ^1^H NMR (300 MHz, DMSO-d6) δ: 7.90 (H7, d, *^3^J_(7-6)_* = 6.9, 1H), 7.72 (H4, d, 1H), 7.40 (H6, t, *^3^J_(6-7)_* = 6.9, 1H), 7.27 (H5, H16, H17, m, 5H), 7.20 (H18, s, 1H), 4.76 (H14, s, 2H). ^13^C NMR (75 MHz, DMSO-d6) δ: 186.9 (C12), 160.8 (C11), 158.1 (C2), 148.71 (C9), 136.6 (C15), 132.2 (C8), 129.3 (C17), 128.6 (C16), 128.4 (C18), 127.5 (C6), 125.3 (C5), 122.7 (C7), 121.7 (C4), 49.0 (C14). Mass analysis ESI [M-H]^+^ (*m*/*z*): 328.0634 found, 328.05 calculated. HPLC purity: 97.6%.

*N-(benzo[d]thiazol-2-yl)-2-((2-methoxybenzyl)amino)-2-thioxoacetamide (***1b***)*. Obtained from 0.343 g of amine **b** and chloroacetamide **1** in 64% yield (0.519 g), m.p. 167–168 °C after recrystallization from ethanol. IR (cm ^−1^) ν: 3336 (m, N-H amide I), 3248 (br, N-H thioamide I), 1692 (s, C=O), 1530 (m, N-H + C-N amide II), 1240 (s, C=S). ^1^H NMR (300 MHz, CDCl_3_) δ: 11.31 (H10, s, 1H), 9.77 (H13, s, 1H), 7.85 (H4, H7, m, *^3^J_(4-5)_* = 7.2, *^4^J_(7-5)_*= 1.2, 2H), 7.48 (H5, ddd, *^3^J_(5-4)_* = 7.2, *^4^J_(5-7)_* 1.2, 1H), 7.35 (H6, H17, H20, m, 3H), 6.95 (H18, H19, m, 2H), 4.89 (H14, d, 2H), 3.91 (H21, s, 3H). ^13^C NMR (75 MHz, CDCl_3_) δ: 182.3 (C12), 157.8 (C16), 156.4 (C2), 156.3 (C11), 149.0 (C9), 132.5 (C8), 130.9 (C15), 130.3 (C20), 126.7 (C5), 124.6 (C6), 122.9 (C18), 121.8 (C7), 121.6 (C4), 120.8 (C19), 110.7 (C17), 55.7 (C21), 46.9 (C14). Mass analysis ESI [M-H]^+^ (*m*/*z*): 358.0779 found, 358.06 calculated. HPLC purity: 99.6%.

*N-(benzo[d]thiazol-2-yl)-2-((4-methoxybenzyl)amino)-2-thioxoacetamide (***1c***)*. Obtained from 0.343 g of amine **c** and chloroacetamide **1** in 43% yield (0.348 g), m.p. 168–170 °C. IR (cm ^−1^) ν: 3301 (m, N-H amide I), 3209 (br, N-H thioamide I), 1676 (s, C=O), 1508 (m, N-H + C-N amide II), 1507 (s, N-H + C-N thioamide II), 1240 (s, C=S). ^1^H NMR (300 MHz, CDCl_3_) δ: 11.27 (H10, s, 1H), 9.48 (H13, s, 1H), 7.85 (H7, H4, dddd, *^3^J_(4-5)_* = 8.5, *^3^J_(7-6)_* = 7.8, *^4^J_(4-6)_* = 1.3, 2H), 7.48 (H5, ddd, *^3^J_(5-6)_* = 8.3, *^4^J_(5-7)_* = 1.3, 1H), 7.35 (H6, ddd, *^3^J_(6-5)_* = 8.3, *^3^J_(6-7)_* = 7.2, *^4^J_(6.4)_* = 1.3, 1H), 7.27 (H16, d, *^3^J_(16-17)_* = 8.3, 2H), 6.89 (H17, d, *^3^J_(17-16)_* = 8.3, 2H), 4.79 (H14, d, *^3^J_(14-13)_* = 5.6, 2H), 3.81 (H19, s, 3H). ^13^C NMR (75 MHz, CDCl_3_) δ: 182.8 (C12), 159.9 (C18), 156.3 (C11), 156.2 (C2), 149.0 (C9), 132.6 (C8), 130.0 (C16), 126.7 (C5), 124.8 (C6), 121.9 (C7), 121.6 (C4), 114.6 (C17), 55.6 (C19), 50.3 (C14). Mass analysis ESI [M-H]^+^ (*m*/*z*): 358.0763 found, 358.06 calculated. HPLC purity: 97.7%. Suitable crystals for X-ray diffraction analysis were obtained from a saturated chloroform solution at R.T.

*N-(benzo[d]thiazol-2-yl)-2-(phenethylamino)-2-thioxoacetamide (***1d***)*. Obtained from 0.303 g of amine **d** and chloroacetamide **1** in 40% yield (0.314 g), m.p. 186–187 °C. IR (cm ^−1^) ν: 3283 (s, N-H amide I), 3215 (w, N-H thioamide I), 1680 (s, C=O), 1508 (m, N-H + C-N amide II), 1526 (s, N-H + C-N thioamide II), 1271 (s, C=S). ^1^H NMR (300 MHz, CDCl_3_) δ: 11.27 (H10, s, 1H), 9.38 (H13, s, 1H), 7.86 (H7, H4, t, *^3^J_(4-5)_* = 6.6, 2H), 7.49 (H5, t, ^3^*J*_(5-6)_ = 7.6, *^3^J_(5-4)_* = 6.6, 1H), 7.36 (H6, H17, t, ^3^*J_(6-5)_* = 7.6, 3H), 7.27 (H18, H19, m, 3H), 4.00 (H14, dt, ^3^*J_(14-15)_* = 7.0, *^3^J_(14-13)_ =* 6.5, 2H), 3.06 (H15, t, ^3^*J_(15-14)_* = 7.0, 2H). ^13^C NMR (75 MHz, CDCl_3_) δ: 183.3 (C12), 156.2 (C11), 156.2 (C2), 149.0 (C9), 137.5 (C16), 132.6 (C8), 129.2 (C17), 128.8 (C18), 127.3 (C19), 126.7 (C5), 124.8 (C6), 121.9 (C7), 121.6 (C4), 47.6 (C14), 33.6 (C15). Mass analysis ESI [M-H]^+^ (*m*/*z*): 342.0813 found, 342.07 calculated. HPLC purity: 97.6%.

*2-(benzylamino)-N-(6-methylbenzo[d]thiazol-2-yl)-2-thioxoacetamide (***2a***)*. It was obtained from 0.268 g of amine **a** and chloroacetamide **2** in 63% yield (0.492 g), m.p. 158–160 °C. IR (cm ^−1^) ν: 3332 (m, N-H amide I), 3270 (m, N-H thioamide I), 1682 (s, C=O), 1547 (s, N-H + C-N amide II), 1512 (s, N-H + C-N thioamide II), 1256 (s, C=S). ^1^H NMR (300 MHz, CDCl_3_) δ: 11.23 (H11, s, 1H), 9.56 (H14, s, 1H), 7.74 (H4, d, ^3^*J_(4-5)_* = 8.3, 1H), 7.62 (H7, dt, ^4^*J_(7-5)_* = 1.8, 1H), 7.37 (H17, H18, H19, m, 5H), 7.31 (H5, dd, ^3^*J_(5-4)_* = 8.3, ^4^*J_(5-7)_* = 1.8, 1H), 4.87 (H15, d, ^3^*J_(15-14)_* = 5.7, 2H), 2.48 (H10, s, 3H). ^13^C NMR (75 MHz, CDCl_3_) δ: 183.2 (C13), 156.1 (C12), 155.4 (C2), 146.9 (C9), 134.9 (C8), 134.7 (C6), 132.7 (C16), 129.2 (C18)), 128.7 (C19), 128.5 (C17), 128.2 (C5), 121.4 (C4), 121.3 (C7), 50.7 (C15), 21.7 (C10). Mass analysis ESI [M-H]^+^ (*m*/*z*): 342.0810 found, 342.07 calculated. HPLC purity: 97.0%.

*2-((2-methoxybenzyl)amino)-N-(6-methylbenzo[d]thiazol-2-yl)-2-thioxoacetamide (***2b***)*. Obtained from 0.343 g of amine **b** and chloroacetamide **2** in 46% yield (0.387 g), m.p. 161–163 °C. IR (cm ^−1^) ν: 3281 (m, N-H amide I), 3247 (w, N-H thioamide I), 1676 (s, C=O), 1519 (s, N-H + C-N amide II and thioamide II), 1243 (s, C=S). ^1^H NMR (300 MHz, CDCl_3_) δ: 11.25 (H11, s, 1H), 9.76 (H14, t, ^3^*J_(14-15)_* = 6.2, 1H), 7.72 (H4, d, ^3^*J_(4-5)_* = 8.3, 1H), 7.60 (H7, dt, ^4^*J_(7-5)_* = 1.7, 1H), 7.33 (H5, H21, dtd, ^3^*J_(5-4)_* = 8.3, ^4^*J_(5-7)_* = 1.7, 2H), 7.25 (H20, dt, 1H), 6.93 (H18, H19, m, 2H), 4.88 (H15, d, ^3^*J_(15-14)_* = 6.2, 2H), 3.89 (H22, s, 3H), 2.46 (H10, s, 3H). ^13^C NMR (75 MHz, CDCl_3_) δ: 182.41 (C13), 157.79 (C17), 156.22 (C12), 155.42 (C2), 146.96 (C9), 134.75 (C6), 132.65 (C8), 130.87 (C21), 130.19 (C20), 128.12 (C5), 122.67 (C16), 121.33 (C4), 121.28 (C7), 120.83 (C19), 110.63 (C18), 55.60 (C22), 46.77 (C15), 21.64 (C10). Mass analysis ESI [M-H]^+^ (*m*/*z*): 372.0914 found, 372.08 calculated. HPLC purity: 98.3%.

*2-((4-methoxybenzyl)amino)-N-(6-methylbenzo[d]thiazol-2-yl)-2-thioxoacetamide (***2c***)*. Obtained from 0.343 g of amine **c** and chloroacetamide **2** in 38% yield (0.322 g), m.p. 167–169 °C. IR (cm ^−1^) ν: 3297 (m, N-H amide I), 3222 (w, N-H thioamide I), 1672 (s, C=O), 1529 (s, N-H + C-N amide II), 1509 (s, N-H + C-N thioamide II), 1249 (s, C=S). ^1^H NMR (300 MHz, CDCl_3_) δ: 11.23 (H11, s, 1H), 9.52 (H14, s, 1H), 7.73 (H4, d, ^3^*J_(4-5)_* = 8.3 Hz, 1H), 7.60 (H7, dt ^4^*J_(7-5)_* = 1.4 Hz, 1H), 7.28 (H5, H17, m, ^3^*J_(17-18)_* = 8.8 Hz, ^3^*J_(5-4)_* = 8.3 Hz, ^4^*J_(5-7)_* = 1.4 Hz, 3H), 6.88 (H18, m, ^3^*J_(18-17)_* = 8.8 Hz, 2H), 4.77 (H15, d, ^3^*J_(15-14)_* = 5.6 Hz, 2H), 3.80 (H20, s, 3H), 2.47 (H10, s, 3H). ^13^C NMR (75 MHz, CDCl_3_) δ: 182.8 (C13), 159.8 (C12), 156.1 (C19), 155.4 (C2), 146.9 (C9), 134.8 (C6), 132.7 (C8), 129.9 (C17), 128.2 (C5), 126.7 (C16), 121.4 (C4), 121.3 (C7), 114.5 (C18), 55.4 (C20), 50.2 (C15), 21.7 (C10). Mass analysis ESI [M-H]^+^ (*m*/*z*): 371.1327 found, 371.08 calculated. HPLC purity: 99.3%.

*N-(6-methylbenzo[d]thiazol-2-yl)-2-(phenethylamino)-2-thioxoacetamide (***2d***)*. Obtained from 0.343 g of amine **d** and chloroacetamide **2** in 51% yield (0.412 g), m.p. 202–203 °C. IR (cm ^−1^) ν: 3290 (m, N-H amide I), 3205 (w, N-H thioamide I), 1676 (m, C=O), 1526 (s, N-H + C-N amide II and thioamide II), 1256 (s, C=S). ^1^H NMR (300 MHz, CDCl_3_) δ: 11.22 (H11, s, 1H), 9.44 (H14, s, 1H), 7.74 (H4, d, ^3^*J_(4-5)_* = 8.3, 1H), 7.60 (H7, dt, ^4^*J_(7-5)_* = 1.6, 1H), 7.35 (H18, m, 2H), 7.31 (H5, m, , ^3^*J_(5-4)_* = 8.3, ^4^*J_(5-7)_* = 1.6, 1H), 7.26 (H19, m, 2H), 7.23 (H20, t, 1H), 3.99 (H15 ,td, ^3^*J_(15-16)_* = 7.4, ^3^*J_(15-14)_* = 6.0, 2H), 3.04 (H16, t, ^3^*J_(16-15)_* = 7.4, 2H), 2.47 (H10, s, 3H). ^13^C NMR (75 MHz, CDCl_3_) δ: 183.4 (C13), 156.1 (C12), 155.4 (C2), 147.0 (C9), 137.5 (C17), 134.9 (C6), 132.7 (C8), 129.1 (C18), 128.7 (C19), 128.2 (C5), 127.2 (C20), 121.4 (C4), 121.3 (C7), 47.5 (C15), 33.5 (C16), 21.7 (C10). Mass analysis ESI [M-H]^+^ (*m*/*z*): 356.0977 found, 356.08 calculated. HPLC purity: 98.7%.

### 3.3. Monocrystal X-Ray Diffraction

Single crystal X-ray diffraction data were collected on an Agilent SuperNova diffractometer equipped with graphite-monochromatic Mo Kα radiation (λ = 0.71073 Å) for compound **1c**. The data collection, cell refinement, and data reduction were accomplished using CrysAlisPro software (v 171.36.20) [77] at 293(2) K. Refinement and structure solution were performed using the SHELXL-2018/3 program [78] of the WINGX (2021.3) package [79] (See Appendix A). Material for publication was prepared with Platon (2023) [80] and Mercury (3.8) [81]. Graph set notation was used for the description of non-covalent interactions [82]. CCDC 2418428 contains the supplementary crystallographic data for this paper.

The crystal structure of compound **1c** was analyzed through the Hirshfeld surface [83] approach. The intermolecular interactions and their reciprocal fingerprints [84,85] were calculated in the CrystalExplorer 21.5 software as previously reported [86,87]. The individual energetic components were obtained by linking the software to the Gaussian 09 package [88] using the B3LYP/6-311G(d,p) theory. Scale factors [89] were included in the calculation, and the percentage contribution of individual components to the stabilization energy was also determined. The results were depicted as frameworks [89,90] in which the E_elect_ (electrostatic), E_disp_ (dispersion), and E_tot_ (total energy) were included in a 3^3^-unit cell with a 3.8 Å radius, a tube size of 180 kJ mol^−1^, and a 10 kJ mol^−1^ cut-off value.

### 3.4. Docking Simulation

Both families of MTA-BTs **1a**–**b**, **1d**, and **2a**–**d** were drawn using ChemSketch (2020-1.2), and the correct connectivity and geometry were confirmed using Gauss View 6.0. For compound **1c**, the crystalline structure solved by X-ray diffraction was employed. The geometry optimization of compounds was performed using Gaussian 09W with a semiempirical method at the AM1 level. The catalytic domain-2 of HDAC6 enzyme, DD2-HDAC6 (PDB: 5G0J), was taken from a previous work [23], and the protein structure was prepared using Autodock Tools (1.5.7). The partial charges for protein and ligand (Kollman and Gasteiger, respectively) were added. The grid box was centered at x, y and z (−55.3669, 62.3515, −1.0206) of 60 Å^3^ with a grid spacing of 0.375 Å^3^. Autodock Vina with BFGS (Broyden-Fletcher-Goldfarb-Shanno) algorithm and hybrid score function (X-Score) was used for a randomized 100-individual population [91,92]. The validation was performed by redocking with Nexturastat A (NA), the crystalized ligand of DD2-HDAC6, and the results were expressed as the free energy of binding (ΔG_b_^o^) (See Appendix A). Docking results were analyzed and processed with Discovery Studio 20.

### 3.5. Antiproliferative Activity Assays

MDA-MB-231 adenocarcinoma cell line was procured from ATCC (CRM-HTB-26TM). MCF-7 (adenocarcinoma), MCF-10A (non tumorigenic breast), and NIH/3T3 (fibroblast) cell lines were generously provided by Dr. José Ruben García-Sánchez (Instituto Politécnico Nacional, Escuela Superior de Medicina). Cell culture was performed in the appropriate medium for each cell line (MCF-7, MDA-MB-231, MCF-10A, and NIH/3T3). MCF-7, MDA-MB-231, and NIH/3T3 cells were grown in Dulbecco’s modified Eagle Medium (DMEM) high-glucose with phenol red, supplemented with fetal bovine serum inactivated by heat (10%) (BioWest, Miami, FL, USA) and antibiotic (100 U/mL penicillin and 100 mg/mL streptomycin). MCF-10A were grown in DMEM-F12 medium, supplemented with 5% of horse serum inactivated by heat (Biowest, Miami, FL, USA), 10 mg/mL of insulin, 20 ng/mL of epidermal growth factor, 500 ng/mL hydrocortisone, and antibiotic (100 U/mL penicillin and 100 mg/mL streptomycin). All cell lines were incubated at 37 °C with 5% CO_2_ after reaching 80% confluence at 24 h. The cells were seeded in a 96-well cell culture plate (10,000 cells/well). After incubation, the cells were treated with compounds (**1a**–**1d** and **2a**–**2d**) at varying concentrations (100, 50, 25, 12.5, and 6.25 µM) and then incubated for 42 h under standard conditions. Cell viability was determined using the MTT [3-(4,5-dimethylthiazol-2-yl)-2,5-diphenyl-tetrazolium bromide (Sigma, St. Louis, MO, USA) assay following reported methods [23], Appendix A.

### 3.6. Wound Closure Assay

The wound closure assay was performed according to the reported methods [23] on MDA-MB-231 cell line. For this purpose, cells (2 × 10^5^) were seeded in complete medium on a 24-well plate and incubated for 24 h under standard conditions. After that, the wound gap was made by scratching the cell monolayer with a 200 µL pipette tip. Cell debris and exhausted medium were discarded, washed with PBS, and the medium was replaced. This was followed by treatment with compound **1c** and methotrexate (MTX) at their IC_50_ concentration (52 µM for **1c** and 28 µM for MTX). The wound closure was monitored by microscopy at 12, 24, and 48 h, and the wound area was measured in duplicate in three independent experiments using ImageJ (1.54k) software. The results were expressed as a percentage of the control (cell culture medium with 0.1% DMSO).

### 3.7. Apoptosis Induction Assays

The annexin V cytometry assay was performed on malignant cells (MCF-7 and MDA-MB-231) and non-malignant cells (NIH/3T3) using the Annexin V-FITC Apoptosis Detection kit (Santa Cruz Biotechnology). The corresponding cells were seeded in a 12-well plate as previously described and incubated under standard conditions (1.5 × 10^5^ cells/well). After 24 h of incubation, the exhausted medium was replaced, and cells were treated at the IC_50_ concentration for compound **1c** and MTX. After 48 h, the exhausted medium was recovered for cytometric measurements, and the cells were trypsinized, inactivated with DMEM, centrifuged, washed with PBS and centrifuged (1500 rpm, 5 min, 4 °C). The PBS was removed, and the remainder was drained on gauze. The formed pellet was shaken with 300 µL of the buffer from the flow cytometry kit (Miltenyi Biotec, Bergisch Gladbach, Germany), which had previously been prepared at a concentration of 1%. The sample was left to incubate for 15 min at room temperature, protected from light, and then 100 µL aliquots were taken to the flow cytometry equipment (BD FACSAria Fusion, Franklin Lakes, NJ, USA). Five microliters of annexin and two microliters of propidium iodide (100 µg/mL) were added to the reaction with the cells. The samples were stored for 15 min at room temperature in the dark, and analysis was performed using cytometry [70].

## 4. Conclusions

In this work, two families of MTA-BT derivatives, **1a**–**d** and **2a**–**d**, were efficiently synthesized using elemental sulfur for thiocarbonyl formation, in good to moderate yields (76–38%). Both families of MTA-BTs were structurally characterized using ^1^H, ^13^C, and 2D NMR, as well as mass spectrometry. The molecular structure of **1c** was confirmed by monocrystal X-ray diffraction. According to OSIRIS, the synthesized compounds have favorable physicochemical, ADME, and toxicological properties. It was demonstrated that the proposed structures are suitable as a drug candidate and have a low risk of presenting toxicological effects. Regarding the biological activity of this group of compounds, MTA-BTs **1a**, **1c**, **2a**, and **2c** exhibited antiproliferative effects at concentrations ranging from 32 to 95 µM. The results were similar to those reported in the literature for HDAC6 inhibitors. The most promising compound was **1c**, which even demonstrated moderate selectivity (1.7) between the tumorigenic MCF-7 and non-tumorigenic human MCF-10A cells. In addition, **1c** can also significantly reduce the cell migration in triple-negative breast cancer cells, which represents a big challenge in the treatment of this aggressive and metastatic cancer. Compound **1c** also increased the number of late apoptotic cells in MCF-7 cells, indicating that the antiproliferative effect is mediated through apoptosis, which is the programmed mechanism of cell death. These results are consistent with in silico docking calculations, which predicted the best affinity of **1c** to the DD2-HDAC6 catalytic site. The X-ray diffraction confirmed the conformation adopted by **1c** in the catalytic site of DD2-HDAC6. Although MTA-BTs do not have superior performance compared to MTX, their activity is similar to that of other HDAC6 inhibitors. Therefore, the results herein reported will undoubtedly contribute to the pursuit of better alternatives in breast cancer treatments.

## Data Availability

The data supporting this article have been included as Appendix A.

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
