# Peer review of "Monothiooxalamide–Benzothiazole Hybrids: Predictive Docking on HDAC6, Synthesis, Molecular Structure, and Antiproliferative Activity on Breast Cancer Cells"

_ijms, 2025, doi:10.3390/ijms26178684_

Round 1
Reviewer 1 Report
Comments and Suggestions for Authors
Dear Editor,
The submitted manuscript "Monothiooxalamide-benzothiazole hybrids: predictive docking on HDAC6, synthesis, molecular structure, and antiproliferative activity on breast cancer cells" presents a detailed study on the synthesis and biological evaluation of monothiooxalamide-benzothiazole derivatives.
The study is comprehensive and methodologically sound, includes synthesis of two families of MTA-BT derivatives, in vitro evaluation of antiproliferative activity, apoptosis induction, molecular docking studies and physicochemical and toxicological properties prediction.
The authors have well described the synthesis and characterization of the synthesized compounds. Also, docking was successfully performed with positive results. These results are predicted the best affinity of 1c to the DD2-HDAC6 catalytic site and consistent with in vitro evaluation of antiproliferative activity. Although the IC50 values of the most potent compound 1c are quite high (>32 mM) in tumor cell lines used for this study, the authors have provided convincing arguments supporting this compound as a potencial anticancer candidate.
Here are some recommendations:
Please check whether in line 97 should contain I and II or 1 and 2.
References 6 and 7 are the same.
I suggest replacing reference 9 related to antimicrobial, and antifungal activity with more appropriate reference (line 47).
I would recommend the publication of the manuscript in the International Journal of Molecular Sciences after minor corrections.
Best regards,
Author Response
CDMX august 25, 2025
Manuscript ijms-3832172
Journal: International Journal of Molecular Sciences
Reviewer 1
Dear Editor,
The submitted manuscript "Monothiooxalamide-benzothiazole hybrids: predictive docking on HDAC6, synthesis, molecular structure, and antiproliferative activity on breast cancer cells" presents a detailed study on the synthesis and biological evaluation of monothiooxalamide-benzothiazole derivatives.
The study is comprehensive and methodologically sound, includes synthesis of two families of MTA-BT derivatives, in vitro evaluation of antiproliferative activity, apoptosis induction, molecular docking studies and physicochemical and toxicological properties prediction.
The authors have well described the synthesis and characterization of the synthesized compounds. Also, docking was successfully performed with positive results. These results are predicted the best affinity of 1c to the DD2-HDAC6 catalytic site and consistent with in vitro evaluation of antiproliferative activity. Although the IC50 values of the most potent compound 1c are quite high (>32 mM) in tumor cell lines used for this study, the authors have provided convincing arguments supporting this compound as a potencial anticancer candidate.
Here are some recommendations:
Comment1. Please check whether in line 97 should contain I and II or 1 and 2.
-The numbers I and II were replaced by the correct numbers 1 and 2.
Comment 2. References 6 and 7 are the same.
-Reference 6 was corrected and reference 7 was replaced by the former reference 9, actually references 10, 11 and 13:
Comment 3. I suggest replacing reference 9 related to antimicrobial, and antifungal activity with more -appropriate reference (line 47).
-The former reference 9 was replaced by:
Tratrat C. Benzothiazole as a Promising Scaffold for the Development of Antifungal Agents. Curr. Top. Med. Chem. 2023, 23, 491-519. doi:org/10.2174/1568026623666230124152429
Comment 4. I would recommend the publication of the manuscript in the International Journal of Molecular Sciences after minor corrections.
-Thank you for your kind recommendation
Reviewer 2 Report
Comments and Suggestions for Authors
This article concern the questions of the synthesis and properties of monothiooxalamide-benzothiazole hybrids. The topic of the paper is interesting and the manuscript is writen well. After some minor revision this article can be accepted fior the publication.
My remarks:
"Nitrogen-sulfur-containing heterocyclic compounds such as thiazole and benzothia-43 zole (BT) are biologically important building blocks in drug discovery and development."
This sentence should be supported by respective examples from the recent litearture.
Scheme 1: the reaction conditions should be presented consequently in the order temperature,solvent,time,yield below the arrow.
Coupling constants should be specified on the (a) part of the Figure 3.
The application of the numerical model for the molecular simulations should be specified.
Similar molecular docking studies for sp2-conjugated systems including carbon and nitrogen atoms were very recently described [10.3390/molecules29215066].
More technical details for HPLC experiments should be added (type of the column, measuring program etc.)
Do products were recrystallised before the measuring of mp's? If yes, the solvent for the crystallisation should be specified in the compound metric.
Author Response
CDMX august 25, 2025
Manuscript ijms-3832172
Journal: International Journal of Molecular Sciences
Reviewer 2
This article concerns the questions of the synthesis and properties of monothiooxalamide-benzothiazole hybrids. The topic of the paper is interesting and the manuscript is writen well. After some minor revisions this article can be accepted for the publication.
-Thank you for your kind recommendation
My remarks:
"Nitrogen-sulfur-containing heterocyclic compounds such as thiazole and benzothia-43 zole (BT) are biologically important building blocks in drug discovery and development."
This sentence should be supported by respective examples from the recent literature.
-The following references were added:
[4] Colorado-Peralta, R.; Olivares-Romero, J.L.; Rosete-Luna, S.; García-Barradas, O.; Reyes-Márquez, V.; Hernández-Romero, D.; Morales-Morales, D. Copper-Coordinated Thiazoles and Benzothiazoles: A Perfect Alliance in the Search for Compounds with Antibacterial and Antifungal Activity. Inorganics 2023, 11, 185. https:// doi.org/10.3390/inorganics11050185.
[5] Bhat, M.; Belagali, S.L. Structural Activity Relationship and Importance of Benzothiazole Derivatives in Medicinal Chemistry: A Comprehensive Review. Mini-Reviews in Organic Chemistry 2023, 17, 323-350. doi: 10.2174/1570193X16666190204111502
[6] Christine Shing Wei Law &Keng Yoon Yeong. Current trends of benzothiazoles in drug discovery: a patent review (2015–2020). Expert Opinion on Therapeutic Patents 2022, 32, 323- 350. Doi:org/10.1080/13543776.2022.2026327
[7] Dawood, D.H.; Anwar, M.M. Recent advances in the therapeutic insights of thiazole scaffolds as acetylcholinesterase inhibitors. Eur. J. Med. Chem. 2025, 287, 117331. doi.org/10.1016/j.ejmech.2025.117331
Scheme 1: the reaction conditions should be presented consequently in the order temperature,solvent,time,yield below the arrow.
-Scheme 1was modified as requested.
Coupling constants should be specified on the (a) part of the Figure 3.
-Coupling constants were included in the footnote of Figure 3:
(a) Spin-spin splitting of H14 protons of compounds 1c (3J(14-13) = 5.6 Hz) and 1d (3J(14-15) = 7.0 Hz, 3J(14-13) = 6.5 Hz), respectively.
The application of the numerical model for the molecular simulations should be specified.
- In the case of docking, the numerical method can be found in lines 535-537.
-Autodock Vina with BFGS (Broyden-Fletcher-Goldfarb-Shanno) algorithm and hybrid score function (X-Score) was used for a randomized 100-individual population [87,88].
Similar molecular docking studies for sp2-conjugated systems including carbon and nitrogen atoms were very recently described [10.3390/molecules29215066].
-Thank you for the reference but I could not find any docking study on it.
More technical details for HPLC experiments should be added (type of the column, measuring program etc.)
-The following data were included: using a ZORBAX XDC-C18 column, at 25 °C.
Do products were recrystallised before the measuring of mp's? If yes, the solvent for the crystallisation should be specified in the compound metric.
-The following lines are included in the 420 and 421:
Compounds 1a and 1b were purified by recrystallization from ethanol, and MTA-BTs by column chromatography using a hexane/ethyl acetate mixture of suitable polarity as eluent.
And:
“After recrystallization from ethanol” in lines 425 and 436.